# Viral Vectors Applied for RNAi-Based Antiviral Therapy

**DOI:** 10.3390/v12090924

**Published:** 2020-08-23

**Authors:** Kenneth Lundstrom

**Affiliations:** PanTherapeutics, CH1095 Lutry, Switzerland; lundstromkenneth@gmail.com

**Keywords:** RNA interference, shRNA, siRNA, miRNA, gene silencing, viral vectors, RNA replicons, COVID-19

## Abstract

RNA interference (RNAi) provides the means for alternative antiviral therapy. Delivery of RNAi in the form of short interfering RNA (siRNA), short hairpin RNA (shRNA) and micro-RNA (miRNA) have demonstrated efficacy in gene silencing for therapeutic applications against viral diseases. Bioinformatics has played an important role in the design of efficient RNAi sequences targeting various pathogenic viruses. However, stability and delivery of RNAi molecules have presented serious obstacles for reaching therapeutic efficacy. For this reason, RNA modifications and formulation of nanoparticles have proven useful for non-viral delivery of RNAi molecules. On the other hand, utilization of viral vectors and particularly self-replicating RNA virus vectors can be considered as an attractive alternative. In this review, examples of antiviral therapy applying RNAi-based approaches in various animal models will be described. Due to the current coronavirus pandemic, a special emphasis will be dedicated to targeting Coronavirus Disease-19 (COVID-19).

## 1. Introduction

Since idoxuridine, the first anti-herpesvirus antiviral drug, reached the market in 1963 more than one hundred antiviral drugs have been formally approved [1]. Despite that, there is a serious need for development of novel, more efficient antiviral therapies, including drugs and vaccines, which has become even more evident all around the world today due to the recent coronavirus pandemic [2]. In this context, many drugs have been subjected to repurposing such as remdesivir, originally developed for Ebola virus disease (EVD) [3], hydroxychloroquine used for treatment of malaria [4] and lopinavir/ritonavir applied for HIV/AIDS therapy and prevention [5]. Unfortunately, follow-up studies have provided discouraging findings in many cases for instance, showing no benefit of hydroxychloroquine treatment of Coronavirus Disease.19 (COVID-19) patients [6]. Furthermore, very recently in search for alternative approaches, expression of 26 severe acute respiratory syndrome coronavirus-2 (SARS-CoV-2) proteins in human cells identified 332 human proteins showing protein–protein interaction by affinity-purification mass spectrometry [7]. Further analysis identified 66 druggable human proteins or host factors, which were targeted by preclinical compounds, drugs in clinical trials or FDA-approved drugs. When a subset was screened in multiple viral assays, two sets of pharmacological agents with antiviral activity were identified, which could support therapeutic interventions of COVID-19. 

Alternative strategies for antiviral drugs are naturally addressed currently including 5-substituted 2′-deoxyuridine analogues, nucleoside and pyrophosphate analogues, reverse transcriptase inhibitors, protease and integrase inhibitors, entry inhibitors, acyclic guanosine and nucleoside phosphonate analogues, immunostimulators, interferons and antimitotic inhibitors [1]. One approach that has received attention recently is the application of RNA interference (RNAi) for viral diseases. In this review are described the mechanisms behind RNAi and gene silencing and how these can potentially be implemented as therapeutic interventions to prevent viral replication and act as antiviral drugs. Examples of virus-based gene silencing in animal models are presented. Due to the current COVID-19 pandemic, a special emphasis is given to the application of gene silencing for prevention of coronavirus replication as a means for antiviral therapy.

## 2. RNAi and Gene Silencing

RNA-based gene silencing was originally discovered in plants and *Caenorabditis elegans* [8]. More recently, gene silencing was confirmed to occur in other organisms [9,10] and more than 6000 micro-RNA (miRNA) sequences have been identified in prokaryotes and eukaryotes [11] with more than 1000 of human origin [12,13]. The mechanism of gene silencing can occur either through mRNA degradation or suppression of translation. In the case of small interfering RNA (siRNA), the multiprotein component RNA induced silencing complex (RISC) binds the double-stranded siRNA, the siRNA strands are separated and the antisense single-stranded siRNA guides RISC to the target mRNA for degradation [14]. In addition to siRNAs, short hairpin RNAs (shRNAs) consist of two complementary 19–22 bp RNA sequences linked by a short loop of 4–11 nucleotides similar to the hairpin naturally present in miRNAs [15]. Delivery of shRNAs can take place by plasmid DNA or viral vectors, and followed by transcription, shRNAs are recognized by the endogenous enzyme Dicer in the cytoplasm, which then processes the shRNAs into siRNAs for binding to target mRNA leading to specific mRNA degradation [16]. Gene silencing is accomplished by miRNAs either through mRNA degradation or suppression of translation [17]. The mechanism of action comprises of transcription of chromosomal miRNA sequences in the nucleus, which are processed to primary miRNAs (pri-miRNAs) by RNAse III endonuclease Drosha and dsRNA-binding protein Pasha, generating the 60–70 nucleotide stem loop precursor miRNA (pre-miRNA). The pre-miRNA is cleaved by the RNase III endonuclease Dicer in the cytoplasm into a 22 nucleotide long ds-miRNA, which then can induce gene silencing. In the case of mRNA degradation, a similar mechanism to siRNA-guided RISC-targeting takes place for miRNAs. In contrast, translation suppression occurs when miRNAs bind to the untranslated region of the mRNA mediated by the RISC. An interesting alternative approach has been to target unusual miRNAs such as miR-451, which are Dicer-independent and instead are processed by the Argonaute 2 (Ago2) enzyme [18]. This approach has allowed therapeutic applications of both non-viral- and viral-based delivery, as described below. 

In comparison to small-molecule or antibody-based drugs, the major advantage of RNAi-based strategies relates to the specific targeting of the drug function to a specific nucleotide sequence [19]. In this context, while small molecule inhibitors generally affect several targets, RNAi can provide highly specific action on unique targets [20]. Whole-genome sequencing and the establishment of comprehensive databases can further expand the potential range for RNAi-based drug discovery [21,22]. However, immune escape and stop codon mutations have been observed in patients with chronic hepatitis B virus (HBV) infections [23], which can be problematic for targeting specific sequences. Moreover, the same problem has been encountered for HIV, leading to the development of anti-escape strategies such as multiplexing, where a large number of shRNA-miRs were expressed using minimal flanking sequences from multiple endogenous miRNAs [24]. 

## 3. RNAi and Bioinformatics

Bioinformatics plays an important role in supporting RNAi-based antiviral therapy. For instance, a stochastic computational model has been designed to aid the development of anti-HIV RNAi therapies [25]. The model simulates the propagation of HIV infection in cells expressing RNAi, displaying targeting of multiple locations in the HIV genome. It also suggests that targeting the HIV transactivating response region (TAR) can be highly successful with a correctly selected RNAi target sequence. The model demonstrates the importance of efficient RNAi delivery systems to avoid leaving reservoirs of cells, where virus can propagate and mutate. Moreover, databases for design and algorithm predictions have been developed for siRNAs [26]. For instance, the viral siRNA database provides detailed siRNA information on 42 human viruses about siRNA sequences, target virus, genes, cell lines and assays as well as siRNA analysis tools such as siTarAlign for the alignment of siRNA and genome sequences of representative viruses [27,28]. Related to HIV, databases provide information on experimentally tested siRNA/shRNA sequences [29], escape mutations and nucleotide mismatches [30] and siRNA efficacy [31]. The design of siRNAs has also been based on analysis of nucleotide frequency at multiple locations in siRNAs [32] and the selection of effective siRNAs directed against conserved regions of HIV, hepatitis C (HCV), influenza virus and SARS-CoV [33]. The Viral siRNA Predictor was developed based on published data using features such as nucleotide frequency, thermodynamic factors and nucleotide location for prediction of siRNA targeting of pathogenic viruses [34]. However, the RNA structure might impose restrictions for RNA therapy based on properties of the HCV genome from primary sequence to tertiary structure [35]. 

## 4. Delivery of RNAi

As with all therapeutic approaches, delivery plays an essential role also for RNAi-based drugs, which has also been reviewed previously [36]. In this context, RNA has always been considered more problematic due its sensitivity to degradation. Moreover, cellular uptake presents another challenge as the cell membrane consists of a negatively charged bilayer of phospholipids and functional proteins, which negatively charged RNAi molecules cannot passively penetrate [19]. For this reason, extensive engineering of lipids and cell-penetrating peptides has enhanced RNA delivery. For instance, liposome-siRNA complexes have been demonstrated to promote transmembrane delivery of RNA [37]. Furthermore, cell-penetrating peptides such as chitosan can increase siRNA uptake and improve its gene silencing effect [38]. It has also been demonstrated that extracellular vesicles can be employed for efficient delivery of cholesterol-conjugated siRNA molecules [39]. Additionally, aptamer-modified extracellular vesicles facilitated tumor targeting of miRNA in tumor-bearing mice [40]. In another approach, aptamer surface labeling of nanoparticles showed selective siRNA delivery to metastatic breast cancer cells [41]. It has also been demonstrated that cholesterol modification resulted in uptake of siRNAs by dendritic cells (DCs) in vitro and in vivo [42]. 

An alternative delivery approach comprises the use of viral vectors. In this context, for instance adenovirus (Ad) [43], adeno-associated virus (AAV) [44] and retrovirus [45] vectors have been applied for gene silencing. Moreover, conditionally replicating lentivirus vectors have been engineered for the expression of siRNA and shRNA sequences [46]. Self-replicating rhabdoviruses [47] and alphaviruses [48] have also been applied for RNAi-based gene silencing. Moreover, it has been proposed that self-replicating RNA viruses could be used as delivery vehicles for small RNAs to induce long-term transcriptional gene silencing (TGS) for therapeutic interventions of various diseases [49]. In this context, cytoplasmic alphaviruses, flaviviruses and Sendai virus (SeV), as well as nucleoplasmic influenza A virus (IAV) and Borna disease viruses (BoDVs) are potential candidates for TGS. 

## 5. Virus-Based RNAi Antiviral Therapy

Currently, there are many viral pathogens, such as HCV, human papilloma virus (HPV), human immunodeficiency virus (HIV), Ebola virus (EBOV), Dengue virus (DENV), and Chikungunya virus (CHIKV) just to name a few, that can be considered a threat to global human health [50]. SARS-CoV-2 responsible for the COVID-19 pandemic is the most serious case in over 100 years [2]. Viral vectors have been applied to some extent for RNAi-based gene silencing targeting viral diseases as listed in Table 1. However, the majority of studies have involved non-viral vectors as described below and summarized in Table 2. 

Adenovirus (Ad) vectors have also been applied for RNAi-based gene silencing, targeting the mosquito-borne flavivirus Tembusu virus (TMUV) [43]. Based on the sequences of the TMUV E and NS5 genes, specifically designed shRNAs expressed from Ad vectors showed efficient downregulation of TMUV RNA replication and virus production in Vero cells. The dose-dependent TMUV virus production lasted for at least 96 h. In another approach, artificial miRNAs (amiRNAs) designed against adenoviral E1A, DNA polymerase and preterminal protein (pTP) mRNA delivered by a replication-deficient Ad vector showed for the pTP amiRNA a 2.6-fold decrease in infectious wildtype Ad [44]. Moreover, combination treatment with Ad-based amiRNA and cidofovir resulted in additive effects leading to a greater than 3-fold reduction in infectious Ad virus progeny. Helper-dependent Ad vectors have also been applied for silencing of HBV replication by delivery of artificial antiviral pri-miRNAs [51]. CMV promoter-based expression of pri-miRNAs from Ad efficiently inhibited HBV replication without causing any toxic effects. Furthermore, introduction of the liver-specific murine transthyretin (MTTR) promoter enhanced the HBV replication knockdown to 94% in mice [52]. Application of trimeric anti-HBV pri-miRs silenced HBV replication for five weeks.

In another approach, the target for antiviral drugs was Ad as immunocompromised patients infected with Ad can develop life-threatening conditions [44]. In this case, amiRs were delivered by self-complementary AAV (scAAV) vectors, which package an inverted repeat genome that can fold into double-stranded DNA without requirement for DNA synthesis or base-pairing between multiple vector genomes [64]. Two scAAV9 vectors carrying three copies of amiR-pTP and three copies of amiR-E1A targeting the Ad5 pTP and E1A genes, respectively, or carrying six copies of amiR-pTP efficiently inhibited Ad5 replication in vitro [44]. Prophylactic administration of scAAV9 to immunosuppressed Syrian hamsters reduced the Ad5 levels by two order of magnitude in the liver and reduced liver damage [44]. In a study on the effect of RNAi on hepatitis B virus (HBV), the anti-HBV effect of AAV7, AAV8 and AAV9 was compared and also the potential of sequential administration of heterologous AAV vectors was studied in transgenic mice [54]. A profound reduction in HBV titers and mRNA and DNA levels was obtained for delivery of HBV-targeted shRNAs from each AAV vector for up to 22 weeks. Cross-administration confirmed that existing AAV8 antibodies completely blocked the anti-HBV RNAi effect of AAV8, but not of AAV7 and AAV9. Sequential administration of AAV8 and AAV9 provided a prolongation of the anti-HBV effect. Moreover, AAV-mediated RNAi therapy showed a significant reduction in hepatic regeneration liver enzymes and liver adenomas [55]. In another study, mono- and tri-meric artificial pri-miRs derived from pri-miR-31 were incorporated into scAAV vectors under the control of a liver-specific modified murine transthyretin promoter [56]. Significant suppression of HBV replication was observed in mice after systemic intravenous administration for at least 32 weeks. In another approach for HBV silencing by AAV vectors, an ancestral AAV (Anc80L65) vector was designed in silico and synthesized in vitro to evade prevailing AAV neutralizing antibodies and can now be characterized in vivo for anti-HBV RNAi activity [65].

Retrovirus vectors have been considered as the prototype for gene therapy, and although providing therapeutic efficacy of X-linked inherited immunodeficiency, unfortunately chromosomal integration induced the activation of the proto-oncogene LIM domain only 2 (LMO2) [66]. For this reason, self-inactivating vector systems carrying an inactivating 3′ long terminal repeat (LTR) deletion transferred to the 5′ LTR during amplification for promoter inactivation has been designed [67]. Although initial RNAi applications of retrovirus-based delivery showed some disappointments, clinical trials have been promising. For instance, Moloney murine leukemia virus (MMLV)-based delivery of anti-HIV-1 tat ribozyme was demonstrated safe in HIV-infected patients in a phase I study [57]. In another phase I clinical trial, a retrovirus vector encoding an anti-HIV ribozyme was expressed in mature hematopoietic cells in HIV patients [58]. The treatment showed detection of vector-containing mature myeloid and T lymphoid cells even in patients with multidrug-resistant infection. In a randomized, double-blind, placebo-controlled phase II study 74 HIV-1 infected adults received a retrovirus-based trans-activator of transcription-viral protein R (tat-vpr)-specific anti-HIV ribozyme (OZ1) [59]. Although the treatment was considered safe and generated biological active ribozyme expression in individuals with HIV, no statistically significant difference in viral load compared to the placebo group was achieved. The lack of efficacy has contributed to shifting the interest from applying retrovirus vectors to lentivirus-based systems. 

RNAi-based gene silencing has been explored using a replicating HIV-based lentivirus vector for the delivery of an antiviral shRNA cassette into HIV-1 susceptible cells with the aim of blocking chronic HIV-1 infection [46]. The engineering of a conditionally replicating doxycycline (dox)-dependent HIV-1 variant allowed efficient virus distribution to HIV-susceptible cells. However, dox withdrawal prevents transcription of the integrated provirus, but the shRNA expression cassette remains active leading to shutdown of HIV-1 replication. Moreover, a lentivirus vector expressing siRNA sequences targeting the HBV polymerase, surface antigen and core protein was evaluated in transduced HepG2 and HepG2 2.2.15 cell lines and human telomerase reverse transcriptase (hTERT)-FH-B fetal human liver cells [60]. The treatment resulted in decreased expression of HBV DNA and RNA with a more pronounced antiviral effect after co-expression of two constructs. In a combinatorial approach, replication-deficient lentivirus vectors were engineered to carry a U6 Pol III promoter-driven shRNA targeting rev and tat mRNAs of HIV-1, a U6 transcribed nucleolar-localizing TAR RNA decoy and a VA1-derived Pol III cassette expressing an anti-CCR5 ribozyme [61]. Each therapeutic RNA targeted a different gene product, and transduction of CD34+ hematopoietic progenitor cells showed superior suppression of HIV replication in comparison to individual shRNAs. The same lentivirus vector was used for the transduction of peripheral blood derived CD34+ hematopoietic progenitor cells and transplantation into four AIDS patients [62]. The transfected cells were successfully engrafted in all four infused patients, and the expression of shRNAs and ribozymes persisted in multiple cell lineages for up to 24 months at low levels. 

In the context of HIV-1-based gene silencing, anti-HIV shRNA and miRNA approaches can be compromised by their potential targeting of HIV-derived sequences in the vector leading to severely impaired vector titers [36,68]. The problem can be addressed by production of shRNA to saturate the RNAi pathway or by application of siRNAs/shRNAs targeting Dicer or Drosha. One solution is to select shRNA sequences, which have been deleted from the lentivirus vector. Moreover, a human codon optimized HIV-1 gag-pol vector can prevent shRNA targeting [69]. Alternatively, vectors based on other types of lentiviruses, such as HIV-2, simian immunodeficiency virus (SIV), feline immunodeficiency virus (FIV) or bovine immunodeficiency virus (BIV), could be utilized. An alternative approach has been to engineer optimized Dicer-independent small RNA duplexes, which are processed by Ago2 [18]. For instance, AgoshRNAs have been designed against HIV-1 [70]. An optimized H1 Pol III promoter with a small five nucleotide loop (CAAGA) for optimal transcription and Ago-2 binding was used. Moreover, as a lentiviral vector was applied, anti-HIV-1 guide sequences were avoided to prevent self-targeting. The AgoshRNAs showed a potent downregulation of CCR5 expression on human T cells and peripheral blood mononuclear cells (PBMCs). Furthermore, it was demonstrated that the CCR5 knockdown provided a significant protection from infection by CCR5-tropic HIV-1 strains [63].

In another strategy, multiplexed miRNA-based shRNAs have been applied for simultaneous suppression of multiple HIV genes [24]. Multiple shRNAs with 30 nucleotides of flanking sequences were inserted in tandem in a lentiviral vector for the expression of one shRNA-miR and six shRNA-miRs targeting CCR5 and the HIV-1 genome, respectively. The lentiviral vector pseudotyped with vesicular stomatitis virus (VSV) G transduced into T cells, significantly inhibited HIV infection in vitro when the construct with the seven shRNA miRs was used, but not when the single shRNA miR was delivered. Moreover, the multiplexed shRNA demonstrated efficient suppression of HIV-1 replication in mice transplanted with human PBMCs. 

In an application of self-replicating RNA viruses, the negative-strand single-stranded (ssRNA) oncolytic VSV was used for the expression of the Nodamura virus protein B2, a known inhibitor of RNAi-mediated immune responses [47]. In this case, VSV expressing B2 generated enhanced viral replication and cytotoxicity, impaired viral genome cleavage and altered miRNA processing in cancer cells. Although in this case, the therapeutic goal was not viral infections as such but the potential of targeting the RNA-mediated antiviral defense in cancer cells. In the context of positive-strand ssRNA viruses, to restrict neuron-specific infection of the oncolytic replication-competent Semliki Forest virus (SFV) vector SFV4, six tandem targets for the neuron-specific miR124 sequences were introduced at the junction between the nonstructural protein genes nsP3 and nsP4 [48]. Intraperitoneal administration of SFV4-miRT124 displayed an attenuated spread into the central nervous system (CNS) of BALB/c mice and resulted in reduced neurovirulence. Moreover, as the SFV4 strain was able to infect and replicate in mouse glioblastoma cells independently of type I interferon defense, the SFV-miRT124 vector with its restricted capacity to replicate in neurons displayed increased oncolytic activity in mouse CT-2A astrocytoma cells and in human glioblastoma cell lines [71]. It was also demonstrated that after a single intraperitoneal injection of C57BL/6 mice transplanted with CT-2A orthotopic gliomas, SFV4-miRT124 was amplified in tumors, leading to significant tumor growth inhibition and prolonged survival. In another study, miRNA sequences for miR124, miR125 and miR134 were introduced into the SFV4 vector, which reduced neurovirulence in the form of neurovirulence in mouse CNS cells [72]. Moreover, a single intravenous injection prolonged survival and provided cure in four of eight mice with NXS2 neuroblastoma xenografts and three of eleven mice bearing CT-2A tumors. However, no cure was observed in mice bearing GL261 glioma tumors.

## 6. RNAi-Based Antiviral Therapy Using Non-Viral Delivery

Although not based on viral vectors, it is appropriate to provide a summary of RNAi-based antiviral drug development utilizing non-viral delivery (Table 2) with a special emphasis on targeting coronaviruses due to the current COVID-19 pandemic (Table 3). In the context of non-viral delivery, cationic lipid-coated gold nanoparticles have been developed for intracellular siRNA delivery [73]. Using this approach, siRNA targeting the open reading frame (ORF) of the HBV X protein showed significant decreases in HBV surface antigen and efficient inhibition of HBV replication on HepG cells. In another study, the human epithelial tumor cell line HEp-2 was persistently infected with the poliovirus Sabin 3 strain followed by transfection with siRNAs targeting both the viral noncoding and coding regions [74]. Repeated treatment resulted in complete cure of persistently infected cell cultures with no escape mutants detected. Related to Marburg virus (MARV) and Ravn virus (RAVN), the therapeutic efficacy of siRNAs targeting the nucleoprotein (NP) and delivered by lipid nanoparticles (LNPs) was evaluated in nonhuman primates at advanced stages of MARV or RAVV disease [75]. Macaques were treated with siRNA-LNPs four or five days after virus infection, which resulted in 100% survival of RAVN-infected animals. In MARV-infected macaques, treatment four days after infection conferred 100% survival, whereas the survival was 50% when siRNA-LNPs were administered five days after infection.

In the case of flaviviruses, the global threat of Dengue virus (DENV) infections has accelerated the development of antiviral drug development, which has included several studies on siRNAs applied for the inhibition of DENV replication [80]. For instance, three siRNAs targeting conserved sequences in the NS4B and NS5 regions of the DENV genome showed efficient inhibition of replication of DENV-1, -3 and -4 in Vero and C6/36 cells [76]. No inhibition was observed for DENV-2. However, a fourth siRNA targeting the NS5 region of DENV-2 provided highly efficient inhibition of DENV-2 replication in both mammalian and insect cells. In an alphavirus study, two siRNAs against conserved regions of the nsP3 and E1 genes of Chikungunya virus (CHIKV) were assessed for virus replication inhibition in Vero cells [77]. Virus titer reductions of 99.6% were detected in siRNA transfected cells. The effect was most prominent at 24 h (99%) with a decrease at 48 h (65%), but with no effect on the expression of house-keeping genes. In another study, four artificial miRNAs (amiRNAs) were designed to target various regions of the CHIKV genome [78]. It was discovered that the amiRNAs significantly inhibited CHIKV replication in Vero cells at both RNA and protein levels, and plaque reduction assays demonstrated a 99.8% inhibition of infectious CHIKV. Moreover, the combination of two amiRNAs against different genomic CHIKV targets showed superior inhibition compared to individual amiRNAs. The inhibitory effect of amiRNAs was also evaluated in combination with antiviral drugs such as chloroquine and ribavirin. The chloroquine–amiRNA combination generated a stronger inhibitory effect than the individual compounds, which could be due to chloroquine targeting the early stage of the CHIKV life cycle, whereas amiRNAs target the late stage of the life cycle. In contrast, the ribavirin–amiRNA combination did not present any improvement in CHIKV inhibition. Moreover, Dicer-independent processing of shRNAs has also been evaluated for the treatment of HCV infections [79]. Synthetic shRNA molecules (sshRNAs) targeting the internal ribosome entry site (IRES) of the HCV genome were encapsulated in LNPs and intravenously administered to HCV-infected immunodeficient mice. The treatment resulted in no hepatocyte toxicity and a specific and durable inhibition of HCV infection.

The current COVID-19 pandemic has made coronaviruses the most urgent targets for development of antiviral drugs, including the exploration of RNAi approaches [2]. Naturally, at this early stage, efforts to apply RNAi against SARS-CoV-2, the virus responsible for the COVID-19 pandemic, have only started. However, RNAi approaches have been applied for SARS-CoV and Middle East respiratory syndrome coronavirus (MERS-CoV) (Table 3). In the case of SARS-CoV, it was demonstrated that siRNAs targeting the S1 and S2 regions of the spike (S) protein expressed from a hairpin cDNA vector efficiently inhibited SARS-CoV replication in Vero E6 cells [81]. In another approach, four chemically synthesized siRNAs targeting the S, nsP-12, -13 and -16 regions showed potent inhibition of SARS-CoV infection and replication in fetal rhesus FRhK4 kidney cells [82], and siRNAs targeting the S and nsP-12 regions also resulted in suppression of SARS symptoms in rhesus macaques in vivo [83]. Moreover, shRNAs targeting the angiotensin converting enzyme-2 (ACE2), the receptor for SARS-CoV, showed silencing of ACE2 expression in Vero cells and reduced infection of SARS-CoV infection [84]. In another approach, the suppression of SARS-CoV entry was addressed by inhibition of the actin-binding protein ezrin by siRNA duplexes [85].

In the context of MERS-CoV, four miRNAs and five siRNAs targeting the ORF1ab region have been rationally designed by computational calculations for gene silencing [86]. RNAi has further been applied for other coronaviruses such as porcine deltacoronavirus (PDCoV), where two plasmid-based shRNAs targeting the PDCoV M and N genes were evaluated in swine testicular (ST) cells [87]. Treatment of ST cells with M and N shRNAs reduced the PDCoV titers by 13.2- and 32.4-fold, respectively, and decreased the viral RNA by 45.8% and 56.1%, respectively. Moreover, plasmid-based shRNAs targeting of the M gene of porcine epidemic diarrhea virus (PEDV) and the swine acute diarrhea syndrome coronavirus (SADS-CoV) inhibited viral RNA expression and impaired virus replication [88]. Finally, related to COVID-19, computational identification of siRNA sequences representative of the SARS-CoV-2 genome and mutation information were carried out [89,90]. Nine potential siRNAs targeting ORF1ab, ORF3a, S, M and N were designed, which will be evaluated for treatment of COVID-19.

## 7. Conclusions

In summary, RNAi-based gene silencing has presented great promise on several levels. Basic research has gained immensely as RNAi has together with proteomics become the most dynamic field of biotechnology [91]. RNAi has also played a prominent role in modern drug screening [92]. The emphasis in this review has been on antiviral therapeutic applications of RNAi, especially highlighting the application of viral vectors for the delivery of siRNAs, shRNAs and miRNAs. The majority of RNAi approaches so far has utilized non-viral delivery methods, which initially suffered from major problems related to RNA delivery and stability [93]. Numerous strategies, including modifications of the 5′ 7-methylguanosine triphosphate (m7 G) Cap structure, the poly (A) tail, 5′ and 3′ end untranslated regions and chemically modified nucleosides have improved the stability of synthetic mRNA molecules [94]. Another approach for improved mRNA delivery comprises the formulation of various nanoparticles based on liposomes, polymers and hybrid lipid polymers, which has provided protection against degradation, enhanced cellular delivery and extended duration of expression in vivo [95]. Although these modifications have improved the delivery of non-viral vectors, the features of superior delivery through highly efficient cell infection and the unmatched expression levels have made the application of viral vectors attractive for RNAi therapy. Viral vectors do not only deliver the RNAi drug, they actually manufacture the drug in the host target cell. Moreover, self-replicating RNA viruses provide extensive RNA amplification directly in the cytoplasm, generating enhanced gene silencing efficacy [96]. Moreover, self-replicating RNA virus vectors can be delivered as naked or encapsulated RNA and DNA plasmids due to their RNA amplification at significantly lower concentrations than synthetic mRNAs or conventional DNA plasmids. However, safety aspects related to the use of viral vectors should not be neglected, and for most viral vector expression systems, vector engineering and appropriate development guarantee safe use in humans. Today, several studies have confirmed the feasibility of RNAi-based gene silencing and inhibition of replication of pathogenic viruses by viral vectors in various animal models (Table 1). Despite recent progress in viral vector delivery, far more studies have been carried out with naked RNAi molecules, DNA-based vectors or liposome/polymer-based nanoparticles (Table 2). It needs to be reminded that translation of the basic concepts of RNAi have faced some limitations and hurdles, which include off-target effects of siRNAs, where ectopically applied siRNAs can alter the expression profiles of several non-targeted transcripts [97]. Moreover, miRNAs can affect down-regulation of target proteins, which in turn can inhibit protein translation and trigger non-specific degradation [98]. Another issue relates to the lethality due to acute liver failure observed in mice after tail vein injection of AAV-based shRNA delivery, which has required to carefully control the levels of ectopic expression of therapeutic shRNAs, as these can be processed into siRNAs eliciting off-target effects and competition with endogenous miRNAs for the RNAi machinery [99]. Another concern comprises the triggering of type I interferon production via activation of Toll-like receptors (TLRs) by certain sequence motifs in siRNAs, which can compromise the RNAi inhibitory effects [100]. This issue has been addressed by including at least one 2’-O-methyl (2’-OMe) in the sense or antisense strand of the siRNA [101].

The current COVID-19 pandemic has also demanded the exploration of all possible alternatives for the discovery of novel or repurposed antiviral drugs for SARS-CoV-2 including RNAi. Finally, it is appropriate to briefly summarize the status of RNAi-based drugs, although they represent other disease indications than viral infections. In this context, RNAi-based treatment of neovascular age-related macular degeneration (Sirna-027) showed a good safety profile in 26 patients in a phase I study [102], but the development was halted when the drug failed to meet a key efficiency endpoint in a phase II clinical trial [103]. Despite this setback and other challenges involving safety and potency, a new era of RNAi-based therapeutics was marked by the FDA approval of the first RNAi-based drug in August 2018 [104]. The drug ONPATTRO™ (patisiran) has also been approved by the European Commission for the treatment of polyneuropathy of hereditary transthyretin-mediated amyloidosis (hATTR) in adults [105]. Although RNAi-based antiviral drugs are still some way off, patisiran may serve as an encouragement for the development of novel and efficient RNAi-based drugs for the treatment of infectious diseases.

## Figures and Tables

**Table 1 viruses-12-00924-t001:** Examples of RNAi-based antiviral therapy using viral vectors.

Indication	Vector/RNAi	Response	Ref
TMUV	Ad5/E and NS5 shRNAs	Inhibition of TMUV in Vero cells	[43]
Ad wt	Ad5/E1A/pTP amiRs	Decrease in Ad wt infection	[44]
HBV	HD Ad/HBV pri-miRs	Inhibition of HBV replication	[52]
HBV	HD Ad MTTR/pri-miRs	Long-term inhibition of HBV replication	[53]
Ad5	scAAV9/pT/E1A amiRs	Inhibition of Ad5 replication in vitro	[44]
Ad5	scAAV9/pT/E1A amiRs	Inhibition of Ad5 replication in hamsters	[44]
HBV	AAV7,8,9/HBV shRNA	Reduced HBV titers, mRNA and DNA levels	[54]
HBV	AAV7,8,9/HBV shRNA	Prevention of HBV hepatocellular adenoma	[55]
HBV	scAAV8/pri-miR-31	HBV suppression for 32 weeks in mice	[56]
HIV	MMLV/tat Ribozyme	Safe delivery in HIV patients in phase I trial	[57]
HIV	MMLV/anti-HIV Ribozyme	MMLV-containing vector in HIV patients	[58]
HIV	MMLV/anti-HIV Ribozyme	Phase II safety, but no efficacy	[59]
HIV	HIV-1/shRNAs	Shutdown of HIV-1 replication	[46]
HBV	HIV/siRNA HBV pol, core	Decrease in HBV DNA and RNA levels	[60]
HIV	HIV/shRNA combination	Suppression of HIV replication	[61]
HIV	HIV/shRNA combination	Persistent expression up to 24 months	[62]
HIV	HIV/AgoshRNAs	Protection against CCR5-tropic HIV-1 strains	[63]
HIV	HIV/shRNA-miRs	Suppression of HIV-1 replication in mice	[24]

AAV, adeno-associated virus: Ad, Adenovirus; AgoshRNAs, Argonaute 2-dependent shRNAs; amiRs, artificial miRNA; HBV, hepatitis B virus; HD Ad, helper-dependent adenovirus; HIV, human immunodeficiency virus; MMLV, Moloney murine leukemia virus; MTTR, murine transthyretin promoter; pri-miRs, primary miRNAs; pTP, preterminal protein; scAAV, self-complementary adeno-associated virus; SFV, Semliki Forest virus; shRNA, short hairpin RNA; TMUV, Tembusu virus.

**Table 2 viruses-12-00924-t002:** Examples of non-viral vector-based RNAi.

Disease	Vector/Target	Effect	Ref
HBV	NP-Gold/siRNA	Inhibition of HBV replication in HepG cells	[73]
Poliovirus	siRNAs	Complete cure in HEp-2 cells	[74]
MARV	LNPs/siRNA	100% survival of MARV infected macaques	[75]
RAVV	LNPs/siRNA	100% survival of RAVV infected macaques	[75]
DENV	NS4B/NS5 siRNAs	Inhibition of DENV replication in cell lines	[76]
CHIKV	nsP3/E1 siRNAs	Titer reduction (99.6%) in Vero cells	[77]
CHIKV	amiRNAs	Inhibition of CHIKV replication in Vero cells	[78]
HCV	LNP/IRES sshRNA	Inhibition of HCV infection	[79]

CHIKV, Chikungunya virus; DENV, Dengue virus; HBV, hepatitis B virus; HCV, hepatitis C virus; IRES, internal ribosome entry site; LNPs, lipid nanoparticles; MARV, Marburg virus; miRNAs, micro RNAs; NP-Gold, gold nanoparticles; RAVV, Ravn virus; siRNAs, short interfering RNAs; sshRNA, synthetic short hairpin RNA.

**Table 3 viruses-12-00924-t003:** Examples of RNAi-based gene silencing against Coronaviruses.

Disease	Vector/Target	Effect	Ref
SARS	Hairpin cDNA/S1S2 siRNAs	Inhibition of replication in Vero E6 cells	[81]
	S, nsP-12/13/16 siRNAs	90% inhibition of replication in FRhK4 cells	[82]
	S, nsP-12 siRNAs	Suppression of SARS symptoms in macaques	[83]
	ACE2 shRNAs	Reduced infection in ACE2-silenced cells	[84]
	Ezrin siRNAs	Knock-down of ezrin	[85]
MERS	ORF1ab siRNAs	Computational predictions for MERS control	[86]
	ORF1ab miRNAs	Computational predictions for MERS control	[86]
PDCoV	M/N shRNAs	Reduced titers and viral RNA in ST cells	[87]
PEDV	M shRNAs	Inhibition of viral RNA and replication	[88]
SADS	M shRNAs	Inhibition of viral RNA and replication	[89]
COVID-19	ORF1b/3a/S,M/N siRNAs	Computational design of siRNAs	[90]

ACE2, angiotensin converting enzyme-2; COVID-19, coronavirus disease; MERS-CoV, Middle East respiratory syndrome coronavirus; miRNAs, micro RNAs; PDCoV, porcine deltacoronavirus; PEDV, porcine epidemic diarrhea virus; SADS-CoV, swine acute diarrhea syndrome coronavirus; SARS-CoV, severe acute respiratory syndrome coronavirus: shRNAs, short hairpin RNAs; siRNAs, short interfering RNAs; ST, swine testicular.

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
