# Peer review of "Viral Vectors Applied for RNAi-Based Antiviral Therapy"

_viruses, 2020, doi:10.3390/v12090924_

Round 1

Reviewer 1 Report

I am happy with the changes made by the author, based on the reviewers' comments.

Author Response

I am happy with the changes made by the author, based on the reviewers' comments.

R: Excellent conclusion!

Reviewer 2 Report

This article is a relatively short summary of RNAi but there is little present of interest to this reviewer. There is little present that isn't well known or hasn't been summarized in recent reviews. Missing are discussions on the limitations and hurdles that must be overcome in order to successfully translate the basic concepts of RNAi to the clinic - and these are major issues. As such the reviewer cannot recommend publication.

Author Response

This article is a relatively short summary of RNAi but there is little present of interest to this reviewer. There is little present that isn't well known or hasn't been summarized in recent reviews. Missing are discussions on the limitations and hurdles that must be overcome in order to successfully translate the basic concepts of RNAi to the clinic - and these are major issues. As such the reviewer cannot recommend publication.

R: A paragraph has been added about the limitation and hurdles of RNAi-based therapy to the Conclusions.

Reviewer 3 Report

This review describes the use of RNAi as an antiviral therapy and summarises results in the literature to date. The review is too broad - the topic of the title "viral vectors" makes up less than half of the text. I understand the author's desire to be thorough, but summarising these other sections that are ultimately off topic would keep the review more focused on the topic at hand. Specifically, section 6 is not a "short summary" as described and in fact has two tables associated with it, compared to one table for the review topic. 

I find the frequent linking of most sections to covid confusing given that this is not the topic of the review. I suggest that rather than being scattered throughout, the author's review/thoughts on the utility of RNAi as potential anti-covid therapy be grouped as a separate section at the end of the review. However, more effort needs to be taken to link this section to viral vectors, as the current format of a review about viral vectors with a large section about non-viral vectors and then non-viral vector approaches to targeting covid is not cohesive.

Minor:

The word "obviously" is added as a precursor to almost all mentions of the significance of the current covid pandemic, which is unnecessary and should be removed.

Although an FDA approved use of these type of therapies is mentioned at the end of the review, this is, as stated, not for the treatment of viral infection and thus i think it is important to emphasise that RNAi use as an anti-viral in humans is still some way off. 

Author Response

This review describes the use of RNAi as an antiviral therapy and summarises results in the literature to date. The review is too broad - the topic of the title "viral vectors" makes up less than half of the text. I understand the author's desire to be thorough, but summarising these other sections that are ultimately off topic would keep the review more focused on the topic at hand. Specifically, section 6 is not a "short summary" as described and in fact has two tables associated with it, compared to one table for the review topic. 

R: I disagree respectfully with the reviewer. The review itself is not very long, in fact only 9 pages for the text including tables. I also think it is good to have a broad description of available viral vector systems. Moreover, I think it is essential to provide a summary on non-viral delivery systems and to satisfy the reviewer, the word “short” has been removed from line 265.

I find the frequent linking of most sections to covid confusing given that this is not the topic of the review. I suggest that rather than being scattered throughout, the author's review/thoughts on the utility of RNAi as potential anti-covid therapy be grouped as a separate section at the end of the review. However, more effort needs to be taken to link this section to viral vectors, as the current format of a review about viral vectors with a large section about non-viral vectors and then non-viral vector approaches to targeting covid is not cohesive.

R: I disagree respectfully with the reviewer. COVID is mentioned in the Abstract once, described in the Introduction (lines 31-34) and mentioned once in both sections 4 and 5. The main description of COVID including Table 3 is presented in section 6 (lines 310-341). I therefore find it surprising that this is confusing as the covid section is at the end before the Conclusions.   

Reviewer 4 Report

This is an excellent review of viral vectors that are being applied in small-RNA-based experimental therapies.  The author present an excellent picture of the current state-of-the-art topics. The article is well-organized and well written. This reviewer has no reservation in recommending the article for publication. 

Author Response

This is an excellent review of viral vectors that are being applied in small-RNA-based experimental therapies.  The author presents an excellent picture of the current state-of-the-art topics. The article is well-organized and well written. This reviewer has no reservation in recommending the article for publication. 

R: Excellent comments!

This manuscript is a resubmission of an earlier submission. The following is a list of the peer review reports and author responses from that submission.

Round 1

Reviewer 1 Report

This single author review article remains unfocused and superficial. Much better and really much more detailed reviews can be found in literature, this one adds little. It also remains unclear what the topic really is: viral vectors for RNAi delivery as the title suggests, targeting of pathogenic viruses, testing in various animal models as the introduction mentions, or a new focus on the COVID-19-causing corona virus. The real focus should be identified and then worked out by providing much more details, including many more citations to the original papers.

Others have published reviews on viral vectors for RNAi delivery, there are many possible references, a complete one is e.g. PMCID:PMC5568016.

Specific issues when RNAi reagents are put into certain vector systems should be discussed, eg HIV-targeting via HIV-based lentiviral vectors causes specific problems that have been worked out in much detail.

Why not mention diverse alternative si/shRNA reagents that have been developed, e.g. Dicer-independent shRNA designs.

The sequence-specificity (2-72) is mentioned as a big advantage, but it is also the reason for virus escape. This has been documented in much detail, especially for HIV. This even led to anti-escape strategies like multiplexing. This simple concept (like drug combination in HIV patients) did work out, but then novel issues popped up (viral titer reduction, recombination on repeat promoter elements etc.). In toto, line this did not work out in clinical terms.

Related to that, whas there really excitement around this topic RECENTLY (1-43)? I think the excitement was 10-15 years ago, now we realize that it did not work out.

1-30: Not sure if we should continue to support Hydroxychloroquine and Lopinavirus as anti-SARS-CoV-2 agents!

The listing of criteria on page 2-90 seems superficial and incomplete, e.g. target RNA structure has been reported to restrict antiviral effects due to stable RNA structure in viral RNA genomes.

Reviewer 2 Report

In this review, the author presents an overview of the current state of the art regarding the use of RNA interference as an antiviral therapeutic. The review focuses on on delivery through viral vectors as an effective way to bring the active molecules, small RNAs, into the cells where they exert their activity, but also provides an overview of some of the literature on non-viral vector delivery. I think the review is generally well-written and can be of interest to the readers of this journal. I do have a few suggestions or remarks though:

In lines 218-219, the author cites a study by Saha et al. but on the use of artificial miRNAs to target CHIKV, but there is no further information provided on the study outcomes. Perhaps provide some more information here.

This is perhaps more a suggestion, but one aspect I was missing a bit in this manuscript is a comparison between vector-based delivery approaches and delivery of naked or nanoparticle-complexed RNAs, since both delivery methods are discussed separately. Some advantages and disadvantages are just briefly mentioned in the Conclusions, but I was wondering whether this could be worked out a bit more in detail, also looking at differences in efficacies and looking at future perspectives and potential remedies for some of the disadvantages of using viral vectors (mainly the safety aspects).

Some smaller language corrections:

Line 44: and how

Line 93: this sentence needs to be looked at. There is something missing after ‘published’

Line 114: remove ‘Among’ or modify the sentence in another way

Line 137: prevents

Line 152: virus (HBV), the anti-HBV effect

Line 180: comma after glioma rather than a period

Line 216: were detected

Line 225: S1 and S1 regions should be S1 and S2 I assume